# Realist evaluation of the role of the Universal Health Coverage Partnership in strengthening policy dialogue for health planning and financing: a protocol

Emilie Robert,[1] Valery Ridde,[2,3] Dheepa Rajan,[4] Omar Sam,[5] Mamadou Dravé,[6] Denis Porignon[4]

For numbered affiliations see end of article.

**Correspondence to**
Dr Emilie Robert;
emilie.robert2@mail.mcgill.ca

## ABSTRACT

**Introduction** In 2011, WHO, the European Union and Luxembourg entered into a collaborative agreement to support policy dialogue for health planning and financing; these were acknowledged as core areas in need of targeted support in countries' quest towards universal health coverage (UHC). Entitled 'Universal Health Coverage Partnership', this intervention is intended to strengthen countries' capacity to develop, negotiate, implement, monitor and evaluate robust and integrated national health policies oriented towards UHC. It is a complex intervention involving a multitude of actors working on a significant number of remarkably diverse activities in different countries.

**Methods and analysis** The researchers will conduct a realist evaluation to answer the following question: How, in what contexts, and triggering what mechanisms, does the Partnership support policy dialogue for health planning and financing towards UHC? A qualitative multiple case study will be undertaken in Togo, Liberia, Democratic Republic of Congo, Cape Verde, Burkina Faso and Niger. Three steps will be implemented: (1) formulating context–mechanism–outcome explanatory propositions to guide data collection, based on expert knowledge and theoretical literature; (2) collecting empirical data through semistructured interviews with key informants and observations of key events, and analysing data; (3) specifying the intervention theory.

**Ethics and dissemination** The primary target audiences are WHO and its partner countries; international and national stakeholders involved in or supporting policy dialogues in the health sector, especially in low-income countries; and researchers with interest in UHC, policy dialogue, evaluation research and/or realist evaluation.

## Strengths and limitations of this study

► The in-depth study of six countries ensures internal validity, and the potential to generalise the findings is increased by building explanations, taking the context into account in the production of outcomes and using existing literature to provide theoretical foundations to the findings.
► Methodological developments are expected, as the study is among the few realist evaluations to focus on policy dialogue, especially in low-income and middle-income countries.
► The study involves training and ongoing supervision of West African researchers who come from a different educational background, contributing to capacity building in health policy and systems research in low-income and middle-income countries.
► The study does not include countries or cases outside of the West African region.
► Comprehensive data collection may be challenging in some countries with an unstable political situation.

## INTRODUCTION

Universal health coverage (UHC) as a core objective within the Sustainable Development Goals is a journey in which multiple stakeholders, from the local, national and international levels, partake. It involves ongoing discussions and negotiations among these stakeholders on the different facets of UHC, among which financing and planning questions.[1 2] Under the leadership of national health authorities, stakeholders will thus have to agree on priorities for action, and health financing and health system organisation modalities.

Policy dialogue is where such discussions take place. WHO describes policy dialogue as an iterative process that targets both the technical and policy aspects of the problem being discussed, involving evidence and sensitive policy discussions, in which a wide range of stakeholders participate.[3] This dialogue has a concrete objective, such as the development of a plan, a strategy or a policy. Policy dialogue is thus understood as a deliberative process by which different stakeholders are brought together to discuss issues of public policy to feed into decision-making.

In low-income and middle-income country (LMIC) settings, the topic area of policy dialogue is attracting interest as a target

intervention for programmes. Policy dialogue is seen as a key governance tool to ensure the coordination and alignment of stakeholders and to develop robust and comprehensive health policies anchored in the UHC concept.[4] Negotiations in the health sector may indeed be challenging due to the heavy influence external aid can have on national priorities, combined with the often limited government steering capacity to counterbalance that influence.[5 6] In the context of UHC, Ministries of Health's convening and brokering role takes on an added layer of importance, as they need to bring together a diverse range of interests to mobilise resources, prioritise interventions and strengthen institutions for UHC.

Following this trend, policy dialogue is becoming a popular research object.[7] In the field of health policy and systems research in LMICs, scholars study policy dialogue as a knowledge translation instrument[8–11] or as a decision-making process. In this case, they notably study the perceptions of what policy dialogue is.[12–14] The actual process of policy dialogue is more rarely investigated.[15 16] In high-income countries, research on policy dialogue as part of decision-making is more diffuse and includes research on research-policy dialogue,[17 18] stakeholder dialogue[19 20] or social dialogue.[21]

In 2011, WHO, the European Union and Luxembourg entered into a collaborative agreement called the *Universal Health Coverage Partnership* to support policy dialogue for health planning and financing as core topic areas of UHC. This Partnership also responds to a resolution from the 64th World Health Assembly calling on WHO to support countries in developing more robust and evidence-based health policies and plans based on an inclusive and participatory process.[22] Implemented in 28 countries, the Partnership is a complex intervention involving a multitude of actors working on a significant number of very diverse activities in different countries. Moreover, it takes place in the policy arena where policy-making is complex and dynamic. This study aims to examine the Partnership through a realist lens, to better understand how and in what contexts it can, through WHO in-country support, contribute to strengthening policy dialogue.

## Realist approach

Traditional evaluation methods face three limitations when studying complex interventions in dynamic contexts, as is the case of the Partnership. First, establishing a causal relationship between the Partnership and its expected outcomes in terms of policy formulation and donors alignment on one hand, and between the Partnership and the indicators usually used in classical evaluations on the other, is onerous as such relationships are not direct and are strongly coloured by the contexts within which WHO intervenes. Second, WHO's role in the Partnership complements its traditional role as technical advisor. Demonstrating how this new approach helps support policy dialogue, and painting a clear picture of the challenges involved, is difficult to do using traditional

evaluation methods. Finally, the Partnership has been implemented in several countries representing a wide variety of contexts, which raises the third limitation of traditional evaluations: the inability to take context into account in explaining an intervention's outcomes and to draw parallels between the experiences of different countries. Therefore, the realist approach was deemed most appropriate as a frame to evaluate the UHC Partnership.

A realist evaluation entails the premise that an intervention does not function on its own and is not, in itself, what produces an outcome.[23] Instead, an outcome is the product of the interaction between a mechanism triggered by the intervention and the context in which the intervention is implemented. A mechanism should be understood as the way in which the actors involved in an intervention react to and reason about the resources made available to them as part of the intervention.[24]

A realist evaluation, therefore, does not attempt to control the variables associated with the implementation context. Rather, it seeks to observe, in the natural environment, the regular—although not systematic—occurrence of an outcome in a particular context, when a mechanism is triggered. These iterations of interactions are referred to as context–mechanism–outcome (CMO) configurations. The objective of a realist evaluation is thus to identify, in the empirical data, iterative interactions between an intervention's context, mechanisms and outcomes. Operationally, it begins with a reflection on the intervention theory and results in the production of an explanatory theory, which is sufficiently abstract to explain how the intervention functions in different contexts. This anchoring in theory requires an explicitly iterative process between the conceptual and theoretical literature underlying the rationale behind the intervention, and the empirical data.

### Intervention theory

Several sources of information were used to develop an initial intervention theory of the Partnership. In addition to reviewing the documents produced by WHO, several meetings were held with stakeholders from WHO headquarters' Department of Health Systems Governance and Financing which implements the Partnership. These meetings aimed to make explicit the logical reasoning underlying the need for and the rationale behind the Partnership. Observations of two annual intercountry meetings bringing together partner countries and WHO teams were also conducted. This process resulted in a generic intervention theory that was discussed at a workshop in Burkina Faso in 2017 and subsequently adjusted to take into account the participants' input (figure 1).

In the frame of the Partnership, WHO supports policy dialogue processes for health planning and health financing. According to WHO, such processes should be inclusive and participatory, fed by knowledge and evidence, and led by the Ministries of Health (MoH). WHO's role is to support MoH to lead structured and transparent policy dialogues. The ultimate goal is to

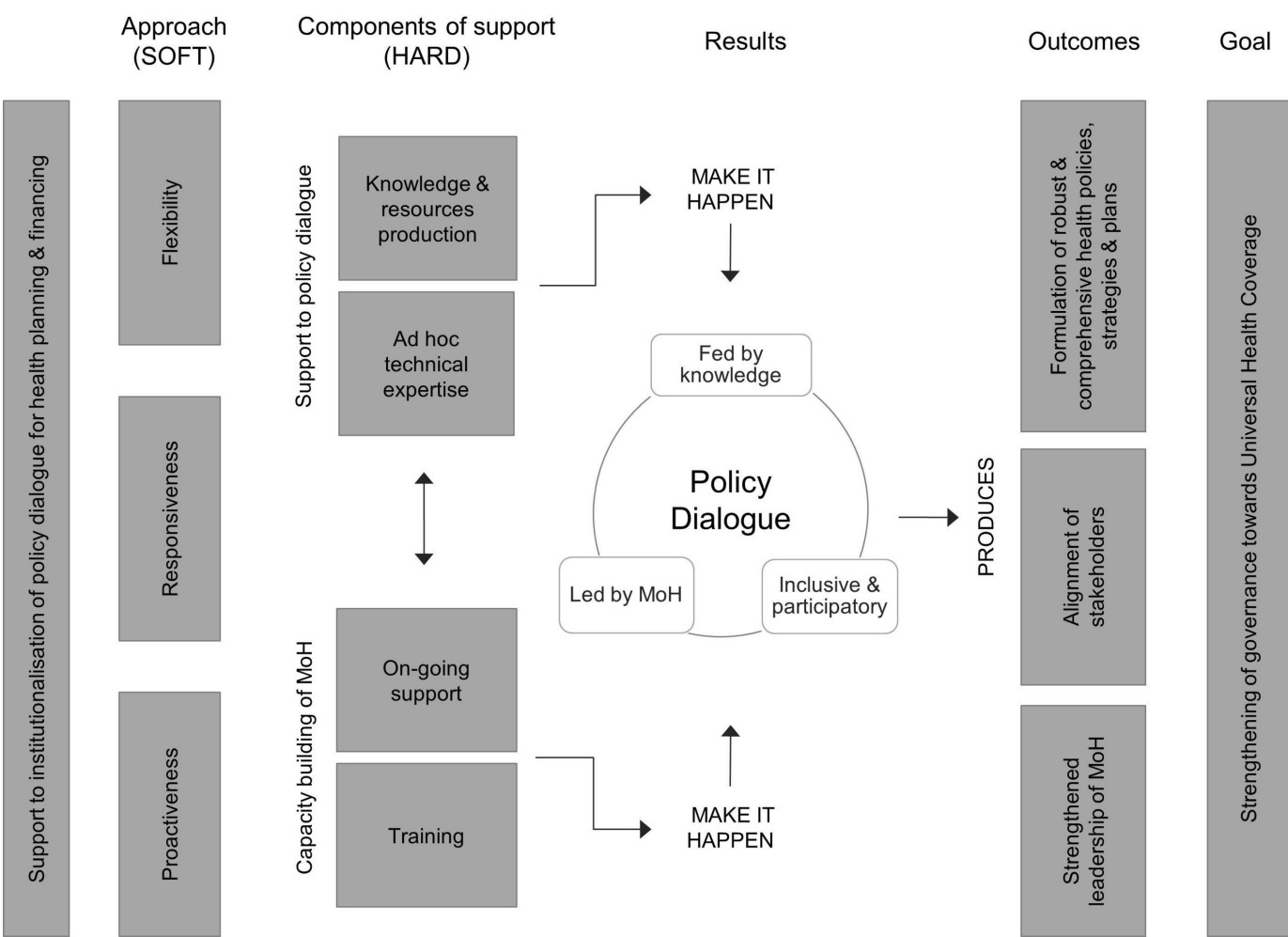

**Figure 1** Generic intervention theory of the Universal Health Coverage Partnership. MoH, Ministries of Health.

contribute to strengthening health systems, especially its governance function, with the assumption that health systems strengthening is the principal means to achieve UHC. WHO support through the Partnership manifests itself through a hard component, including financial resources for a set of activities as well as human resources, and a soft component, which refers to a specific approach to supporting the policy dialogue. The hard component of WHO support includes strengthening of the capacities of MoH to conduct policy dialogue processes. This involves WHO country office ongoing support as well as training. It also involves strengthening of the quality of the policy dialogue itself with the generation of relevant evidence, data and normative guidelines, as well as ad hoc expertise. With regard to WHO's approach, three characteristics are essential: flexibility, that is, the ability to adapt to the changing priorities of the MoH; responsiveness, or the ability to position and respond to requests from the MoH; and proactivity, that is, the ability to anticipate implementation steps and expectations or needs of the MoH.

The resulting policy dialogue is expected to enable the formulation of robust and comprehensive health policies, strategies and plans that promote the principles of UHC such as equity, financial risk protection and quality of care. It should also facilitate the alignment of multiple stakeholders, be they donors, civil society or related ministries. Thus, the Partnership is supposed to contribute to upholding the principles of the Paris Declaration to ensure effective development cooperation. Finally, supporting MoH in the organisation of such dialogues is expected to strengthen the capacities and leadership position of the MoH. To summarise, WHO has three principal roles within the Partnership: convener of policy dialogue processes, policy broker and a relatively traditional role of advisor to the MoH.

WHO may act as a 'boundary spanner', that is, an actor who brings together partners with common interests and goals (such as UHC), while striving to establish a climate of trust, optimism and perseverance.[25] The concept of policy broker is similar to that of boundary spanner. The role of a broker is to bring together groups of actors whose interests and beliefs differ about the issue of concern in the policy dialogue.[26] The broker's objective is to promote consensus in a climate of trust, to stabilise the policy dialogue and to find workable compromises. However, his role is not necessarily a disinterested one, and he might be pursuing a strategic objective, which would

bring him closer to the role of a policy entrepreneur. A policy entrepreneur has an explicit strategic objective, and proposes ideas and stimulates opportunities in ways that are intended to create a demand for the solution he is proposing.[26] These roles are, however, ideal types positioned at two ends of a continuum.[26] Depending on the contexts in which it intervenes, the processes in which it is involved and the issues being discussed, WHO may be called on to play one or another of these various roles.

## METHODS AND ANALYSIS
### Research questions
The question that will guide the study is: How, in what contexts, and triggering what mechanisms, does the Partnership support policy dialogue for health planning and health financing towards UHC, and with what outcomes?

Four specific objectives have been established to respond to this research question:
1. To highlight the contexts in which the Partnership, through WHO support, can, or cannot:
   a. act as a broker, promoting the dissemination and use of data and evidence in policy dialogue and seeking workable compromises;
   b. act as a convener, creating synergy among the actors involved in policy dialogue for health planning and financing;
   c. play its role as an advisor, ensuring transparent and structured policy dialogue in accordance with available evidence and equity principles; and supporting MoH in its leadership and stewardship functions.
2. To identify the mechanisms at work and the outcomes—direct or indirect, expected or not—produced;
3. To clarify the process chains and the links between them, especially highlighting feedback loops;
4. To specify how, in what contexts, triggering what mechanisms and with what outcomes the Partnership, through WHO support, enhances policy dialogue for health planning and financing towards UHC.

### Study design
A realist multiple case study will be undertaken.[27] Table 1 presents a summary of the proposed design. The countries in which the study will be conducted are Togo,

| Table 1 | Summary of the study design |
|---|---|
| Epistemological foundation | Critical realism |
| Design | Multiple case studies |
| Cases | Countries involved in the Partnership |
| Units of analysis | Policy dialogue processes supported by WHO as part of the country roadmaps |
| Type of sample | Contrasted |
| Case sampling strategy | Purposive sampling |

Liberia, Democratic Republic of Congo, Cape Verde, Burkina Faso and Niger. Due to feasibility concerns, only partner countries of the African continent were selected. They were sampled using a non-probabilistic approach. The objective was to select cases—or countries—purposively, so as to obtain a diversity and depth of information on the Partnership. This is therefore a contrasted sample. When selecting cases, purposive sampling involves establishing selection criteria that ensure contrasting cases. The criteria are based on theoretical assumptions about the influence they may have on the Partnership and policy dialogue. An initial list was shared with WHO in order to select the criteria that could be informed by their experience. Based on their country knowledge and experience, they then suggested countries matching the criteria, based on their country knowledge and experience (table 2).

Each country has a roadmap for the Partnership work, in line with its needs. In this study, the unit of analysis is a policy dialogue process defined in the country's roadmap. One process (unit of analysis) will be observed in each country (case). Each process was selected in collaboration with the country teams, taking into account ongoing support at the start of data collection, as well as opportunities for observation. In Burkina Faso and DRC, the processes under study are the policy dialogues around the national health financing strategy, whereas in Cape Verde, Niger and Togo, there are the policy dialogues for the elaboration of the national or regional health plans. In Liberia, the process under study is the elaboration of the IHP+ (International Health Partnership) Compact, which is an aid coordination agreement among health sector stakeholders. The process selected determines the research subquestions to be addressed in each case, the documentation to be obtained, the informants to interview, and the topics to be covered in interviews.

### Study steps
#### Step 1: formulating CMO propositions
Within a realist perspective, researchers raise the questions of the mechanisms, that is, how actors react to the new resources made available through an intervention, and of the contexts that trigger or hinder these mechanisms. The initial intervention theory was divided into two subtheories to clarify what mechanisms and contexts may interact (figure 2 and figure 3). Contexts and mechanisms derive from three sources: literature on partnership synergy[25 28 29]; a scoping study on policy dialogue,[30] looking into knowledge translation, policy change and development studies literature; and interviews conducted with international experts involved in the Partnership. We describe below both subtheories, highlighting mechanisms (M), outcomes (O) and potential contexts (C).

The first subtheory focuses on the Partnership and its expected outcomes (figure 2). It posits that capacity building, through training, technical expertise and ongoing support from WHO, would empower MoH (M), while triggering a shared understanding of governance

**Table 2**  Case selection criteria and reason for sampling

| Criteria* | Togo | Liberia | DRC | Cape Verde | Burkina Faso | Niger |
|---|---|---|---|---|---|---|
| Duration of WHO support | 4 | 4 | 3 | 3 | 3 | 3 |
| Political stability (0=instable; 4=stable) | 3 | 3 | 2 | 3 | 3 | 1.5 |
| Ministerial stability (0=instable; 4=stable) | 3 | 3 | 3 | 2 | 2 | 1 |
| Openness to policy dialogue (0=not open; 4=open) | 3 | 3 | 3 | 2 | 2 | 1.5 |
| Ministry of Health leadership capacity (0=weak; 4=strong) | 3 | 3 | 3 | 3 | 3 | 1.5 |
| Implementation of the national roadmap (0=weak; 4=good) | 3 | 3 | 3 | 2 | 3 | 1 |
| Presence of entry points | Yes | Yes | Yes | Yes | Yes | Yes |
| Additional observations | | Ebola epidemic | Devolution | Archipelago—lower middle-income country | | |

Criteria were noted on a scale from 0 to 4, where 0 is the lowest and 4 the highest. For example, a score of 4 for Togo for criteria 'Duration of WHO support' means that the country benefited from WHO support for the longest period of time.
*At the time of case selection.

and policy dialogue (M). This should result in MoH leading inclusive and participatory policy dialogues (O). Furthermore, the ongoing support provided to MoH, combined with WHO approach (ie, proactiveness, responsiveness, flexibility), should prompt mutual trust (M), which in turn should strengthen their collaboration for policy dialogues (O). Finally, the subtheory postulates that the availability of evidence and data provided through the Partnership would bring about a shared understanding of the needs and policy options by MoH

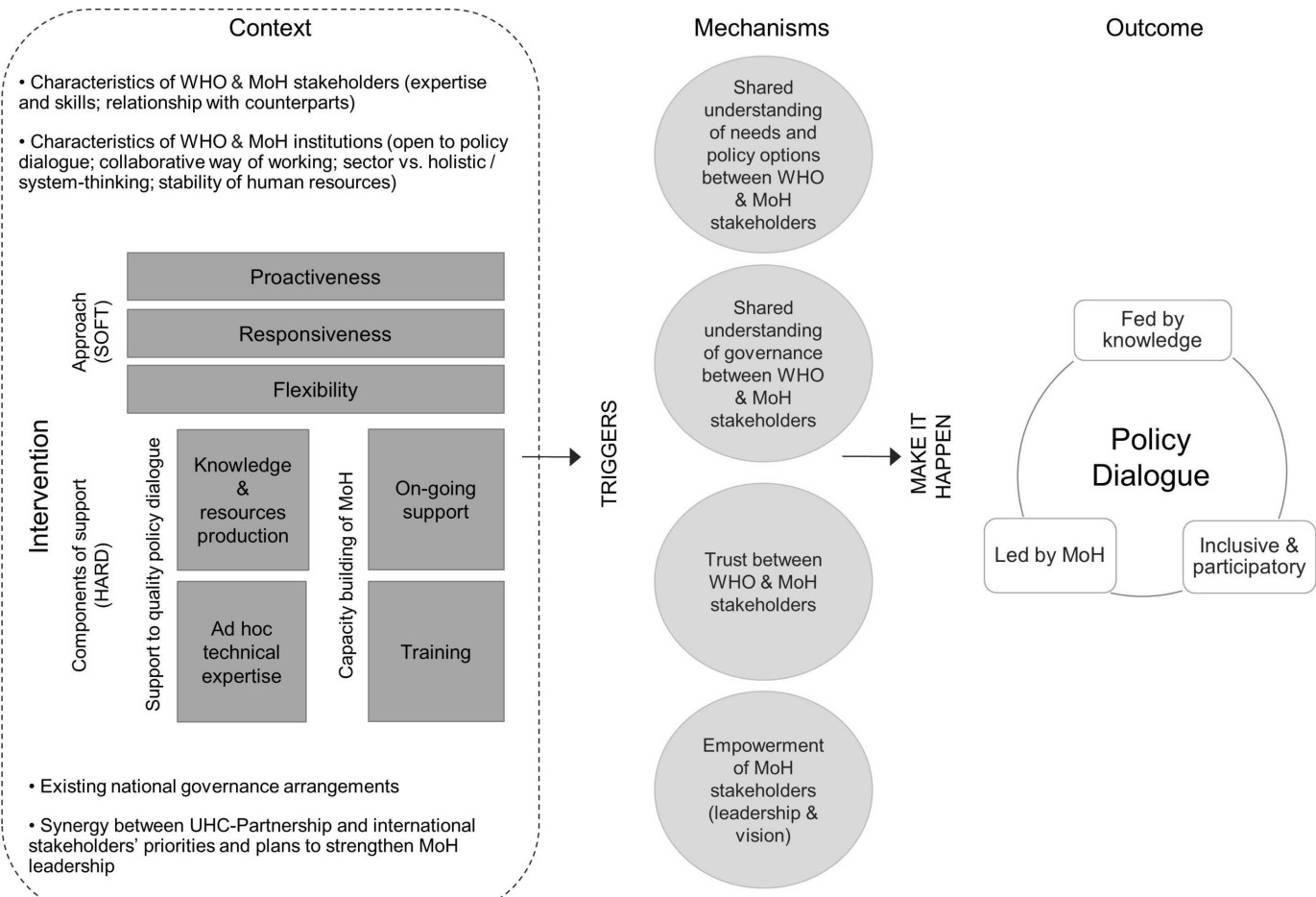

**Figure 2**  Initial subtheory of the Universal Health Coverage Partnership (subtheory 1). MoH, Ministries of Health.

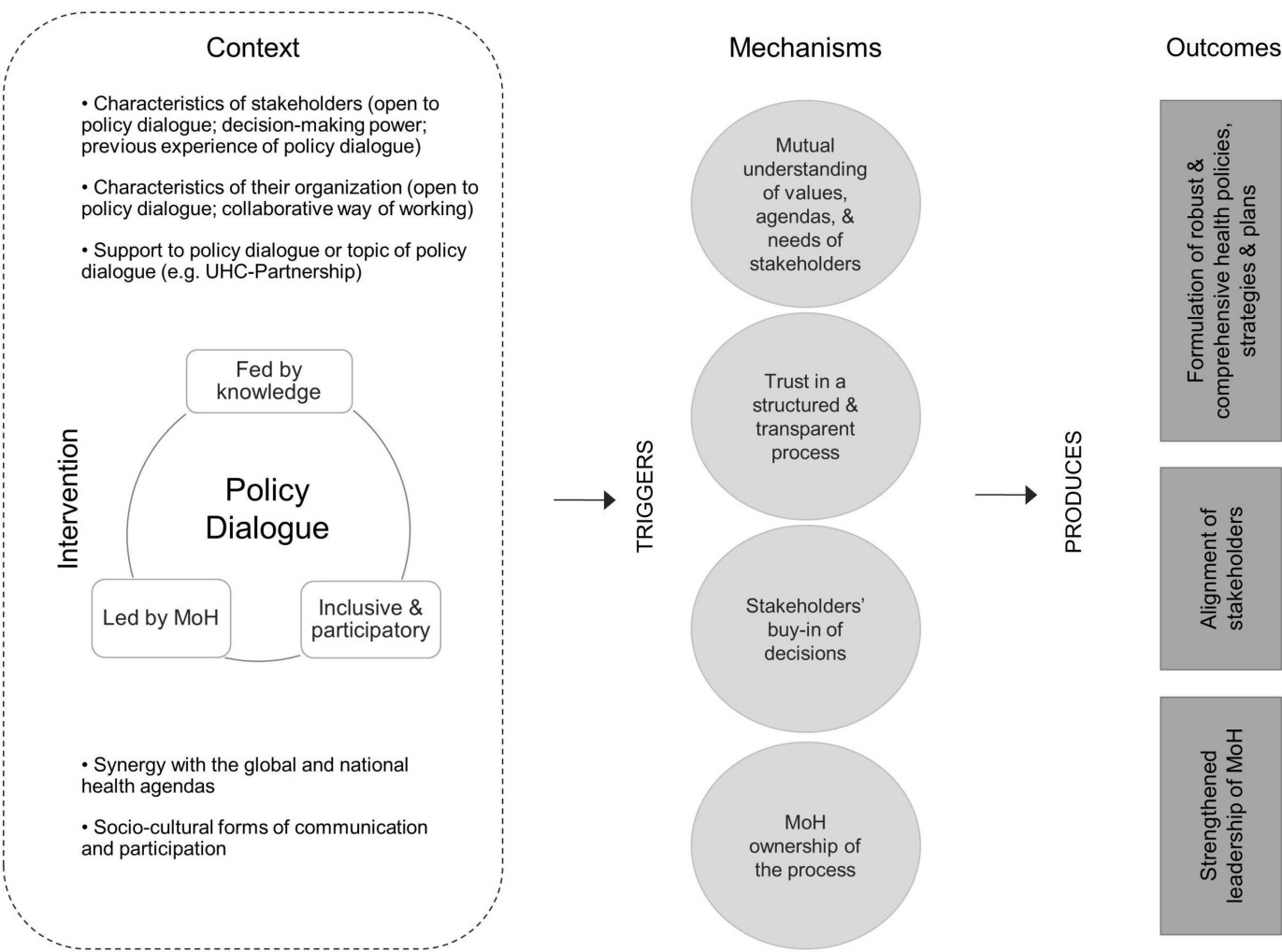

**Figure 3** Initial subtheory of policy dialogue (subtheory 2). MoH, Ministries of Health.

and WHO (M), contributing to an evidence-informed policy dialogue (O). In subtheory 1, mechanisms may be triggered by the Partnership when existing national governance arrangements facilitate the organisation of policy dialogue (C); WHO and MoH have enduring relationships (C); they have the expected expertise and skills (C); they both endorse collaborative and cross-sectoral ways of working (C); human resources from both institutions who are involved in the Partnership are stable (C); and there is synergy between the Partnership and international stakeholders' priorities and plans to strengthen MoH leadership (C).

The second subtheory focuses on policy dialogue and its expected outcomes (figure 3). It posits that an evidence-informed policy dialogue would first trigger a shared understanding among stakeholders of the challenges and possible ways of action (M). This should contribute to more robust and comprehensive health policies, strategies and plans (O) through the formulation of commonly agreed workable propositions. An inclusive and participatory policy dialogue should also prompt a shared understanding of stakes, values and interests of the various actors involved (M), as well as buy-in of the decisions (M), strengthening

stakeholders' alignment on national health priorities (O). Finally, brought under the responsibility of MoH, it should foster MoH sense of ownership of the process (M), contributing to strengthening its leadership (O). Such an outcome may also be the consequence of stakeholders' trust in a structured and transparent policy dialogue process (M). In subtheory 2, policy dialogue may trigger such mechanisms when participants show openness to policy dialogue (C); they have the capacity and legitimacy to actively participate (C); they have previous positive experience of policy dialogue (C); their organisations' culture value collaboration and cross-sectoral ways of working (C); policy dialogue benefit from financial and technical support (eg, the Partnership) (C); there is synergy between the global and national health agendas (C); and policy dialogue matches sociocultural forms of communication and participation to governance and formal events (C).

For each case, we will formulate at least one CMO proposition for each outcome, based on these theories. Additional theoretical literature may be used to fill in potential knowledge gaps, such as capacity building literature and theories about organisational change.

## Step 2: collecting and analysing empirical data

Once the propositions are developed, an interview guide will be drafted for each case. The interviewee is there to help the researcher 'refine' the theory, as each interviewee holds a piece of the complex puzzle of the intervention.[23] Consequently, the interview guide will be adjusted to the interviewee. For example, it is expected that WHO informants will be knowledgeable about the Partnership and its outcomes (subtheory 1), while other stakeholders will be able to share their experience as policy dialogue participants (subtheory 2).

Three sources of information will be used. First, semistructured interviews will be conducted with key informants who will be jointly identified by the researchers and WHO. An initial list will be compiled by WHO and then supplemented using a snowballing technique. The aim is to meet with as many as possible of the participants of the policy dialogue and avoid investigating merely the positive side of the coin, and thus possible bias. Depending on informants' availability, interviews will be conducted in person or by telephone. The interviews will be audio-recorded, except when not allowed by the participants. They will also be recorded in notes taken systematically by the researchers. Recordings will subsequently be transcribed. Between 20 and 30 informants will be interviewed for each case. The principle of empirical saturation will be applied to determine the best time to bring the data collection to a close.[31]

Observations will also be conducted of activities organised either directly by WHO or with WHO collaboration within the Partnership context. These observations will be described and recorded in the logbooks of researchers. Also, all documentation related to the Partnership (eg, national roadmaps, project documents, annual reports) will be reviewed.

Furthermore, to gain a deeper understanding of the national health policy contexts, and particularly of the interests, power relationships and positioning of the actors involved in policy dialogue, a stakeholder analysis will be performed for each case.[32] First, we will compile a list of stakeholders involved in policy dialogue for each case. Then, for each stakeholder, we will attempt to describe their power of action and influence, as well as their interest and involvement with respect to UHC and policy dialogue. The last step will be to analyse how these stakeholders react to the roles played by WHO in policy dialogue and how their reactions influence these two processes. This exercise will be conducted before the interviews in collaboration with the WHO teams, who have first-hand knowledge of the field. The analysis will be completed during the interviews with the stakeholders included on the list.

Using an iterative approach during data collection, we will deepen our understanding of the intervention theory, including 'hypotheses about their subjects' reasoning within a wider model of their causes and consequences' (p. 163).[23] Data analysis will contribute to deepening our understanding further. It will start with a thorough description of the barriers and facilitators for each piece of the intervention theory. We will then move to identify in the data the interaction between context, mechanisms and outcomes. Finally, we will attempt to identify—across interviews—patterns of such interactions, called CMO configurations. Both barriers and facilitators and CMO configurations will be identified in the data through content analysis performed using NVivo software. For the latter, we will (1) identify the outcomes (description), (2) determine the contextual components related to the outcomes (resolution), (3) reconstruct the links between context and outcomes theoretically (abduction), and (4) identify the mechanisms (retroduction).[33] Retroduction aims to construct the theoretical explanation of a phenomenon.[34 35] More specifically, this retrospective reasoning allows moving from describing the outcome to describing what produced that outcome and the context within which

| Table 3 | Summary of the study process | | |
|---|---|---|---|
| **Steps** | **Data sources** | **Methods** | **Status** |
| Building a generic intervention theory | ► Programme documents<br>► Meetings with stakeholders from WHO headquarters' Department of Health Systems Governance and Financing<br>► Observations of annual intercountry meetings (n=2) | Iterative co-building process | Completed (2016) |
| Formulating CMO explanatory propositions | ► Literature review on policy dialogue<br>► Semistructured interviews with experts at WHO headquarters and at regional level (n=13) | ► Scoping study<br>► Content analysis | Ongoing (2017–2018) |
| Identifying CMO configurations | ► Semistructured interviews with WHO national experts, MoH counterparts, participants of policy dialogues (eg, civil society, international stakeholders, connected Ministries and public institutions) in each country<br>► Observations of policy dialogues | ► Stakeholder analysis<br>► Content analysis | 2018 |
| Specifying intervention theory | ► CMO configurations from cases | Transversal analysis | 2019 |

CMO, context–mechanism–outcome; MoH, Ministries of Health.

it takes place,[36 37] via the mechanisms.[38] In case of missing pieces of evidence, additional interviews will be conducted whenever possible.

### Step 3: specifying the theory

In order to specify the intervention theory across cases, a transversal analysis will be conducted using CMO configurations as primary data. Such configurations may be incomplete, with the mechanism generally not being explicit. The missing links will thus be inferred from the theoretical literature, which provides clues to explain the links between causes and outcomes, following a process of intellectual reasoning based on two questions: How can we explain the mechanism in this context and considering this outcome? Is this interpretation supported by the theoretical literature?

This inference takes the form of a demi-regularity when the configuration re-appears several times. A demi-regularity is a regular—although not systematic—occurrence in the interactions among the context, the mechanism(s) and the outcomes.[39] When a configuration is identified only once but its interpretation is supported by the theoretical literature, it is also taken into account, on the condition that it helps to clarify how and under what circumstances the Partnership contributes to each outcome. Each demi-regularity will provide one or more piece of information that, when combined, present a more complete and precise picture of the Partnership. Such an analysis will contribute to specify the explanatory realist theory of the Partnership. Table 3 summarises the study process, including sources of data, methods and status. The research should be completed by mid-2019.

### Patients and public involvement

There is no patient or public involvement in this study.

### ETHICS AND DISSEMINATION
### Protection of key informants

The research objectives and a consent form will be sent by email to informants before the interviews. The rights of informants include the right to refuse to participate in the interview, to end their participation in the study at any time and to not allow their statements to be audio-recorded. Informed consent will be sought in writing from informants. We expect that all informants will be literate, given that they occupy positions of responsibility.

Informants' statements will be kept confidential. The data collected will be kept anonymous at the time of transcription, that is, informants' names will be concealed, as well as their title and their function. Transcriptionists and everyone involved in data collection and analysis will sign confidentiality agreements. A copy of the data will be maintained on a password-protected virtual platform. The data will be destroyed 18 months after the signing of the consent forms.

### Importance of the study

First, in providing lessons on how the Partnership works and what proximal and unexpected outcomes it produces, the research should allow WHO and Partnership donors to improve the Partnership content, to adapt it better to the contexts in which it is implemented, and to tailor the nature of the policy dialogue support to take into account the diverse influences that foster or undermine productive collaboration among stakeholders. This more constructive collaboration should in turn contribute to more robust and actionable health policies, as a means to strengthen health systems so that they can deliver the quality and affordable health services users are entitled to under the banner of UHC.

Second, the realist evaluation has been little used to study interventions on an international scale. Thus, this study will be a unique example in health policy and systems research, and will offer important methodological lessons. Furthermore, it entails the training and ongoing supervision of a team of West African researchers coming from various educational backgrounds and with different expertise. This project will help strengthen capacities in health policy and systems research in LMICs.

Finally, in terms of the topics addressed, the nature of the knowledge produced will be unique and will provide a deeper understanding of the issues associated with interventions aimed to support policy dialogue. As such, it adds to the findings on policy dialogue reported in the *BMC Health Services Research* special issue.[40]

### Dissemination of results

The results of the study will be disseminated on several levels: WHO, including the headquarters, the regional teams and the country teams; WHO's national and international partners, particularly MoH and donors; and the scientific community. With regards to the scientific community, empirical findings and methodological issues will be shared in related symposia and in peer-reviewed journals. Any intention to publish the findings must be approved by WHO before submitting to the journal.

**Author affiliations**

[1]Training and Research Transcultural Team, Research Institute of the McGill University Health Centre, Montreal, Quebec, Canada
[2]Centre Population et Développement, Paris, Île-de-France, France
[3]Institut de recherche en santé publique, Université de Montréal, Montreal, Quebec, Canada
[4]Department of Health Systems Governance and Financing, World Health Organization, Geneva, Switzerland
[5]Inter-Country Support Team, World Health Organization, Ouagadougou, Burkina Faso
[6]Country Office, World Health Organization, Lomé, Togo

**Acknowledgements** We would like to thank the following collaborators at WHO: Juliet Nabyonga, Martin Ota and Kira Koch. We would also like to express our gratitude to the WHO country office and Sébastien Segniagbeto in Togo for their support during the pilot mission, as well as researchers in countries for support with national ethics committees. We finally would like to acknowledge Donna Riley for the translation.

**Contributors** ER and VR designed the study. DR, OS, MD and DP provided critical data and information regarding the intervention, the sampling of cases and

feasibility of the research. ER drafted the first version of the protocol. All authors critically revised the study protocol for important intellectual content and approved the manuscript.

**Funding** This work and the Open Access fee are funded by the Department of Health Systems Governance and Financing of the World Health Organization.

**Disclaimer** WHO as the funder of this research will not have any role in data collection and data analysis. However, any publication will require WHO approval.

**Competing interests** ER and VR are funded as research consultants by the Department of Health Systems Governance and Financing of WHO. DR, OS, MD and DP are experts at WHO and involved within the Universal Health Coverage Partnership.

**Patient consent for publication** Not required.

**Ethics approval** WHO-AFRO ethics review committee (AFR/ERC/2017/02.08) as well as the Ethics Committee of each country included in the study.

**Provenance and peer review** Not commissioned; externally peer reviewed.

**Data sharing statement** Not applicable.

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
