## [Reviewer comments · BMJ Open]

ARTICLE DETAILS

TITLE (PROVISIONAL)	Realist evaluation of the role of the Universal Health Coverage Partnership in strengthening policy dialogue for health planning and financing: A protocol
AUTHORS	Robert, Emilie; Ridde, Valery; Rajan, Dheepa; Sam, Omar; Dravé, Mamadou; Porignon, Denis

VERSION 1 – REVIEW

REVIEWER	Gill Schierhout The Kirby Institute, University of New South Wales, Sydney, Australia
REVIEW RETURNED	19-Apr-2018

GENERAL COMMENTS	General comments This is an ambitious and important study of the implementation of a large systems-based initiative. I have a reservation in the way the realist approach is described here, and think this needs some clarification in the manuscript. I have two overall recommendations that I respectfully suggest that the authors consider. 1. Would the authors please consider presenting the initial program theory in a CMO structure. This will align better with the realist framework as the approach underpinning this study. Currently it is not clear to this reader what the CMO configurations are that will be tested and refined. If it is not possible to do this, I would suggest that the research being conducted is 'theory development' rather than 'theory testing' (see point 4 below).2. I think the authors need to consider how the initial program theory will guide what kinds of data will be collected in the study. I wondered if the initial program theory as presented will be sufficient as a basis for a robust realist evaluation of a complex intervention in complex settings. If it is not, there needs to be a step in the study to develop such a program theory, which then forms the basis for the design of the evaluation. Currently it is not clear how the CMO configurations will emerge from the study.3. My understanding of realist evaluation is that it needs to build on learnings from previous programs that have used similar mechanisms (in possibly unrelated fields). This, in my understanding, the contribution of realism, but I don't see where this is included in the methodology. For example, theory from organisational change, or other disciplines.4. I strongly recommend that the authors please consider revising the goal of the paper to be about "developing a realist theory of change" rather than actually doing a realist evaluation. This is consistent with what the authors own statement about the intended contribution of their analysis, being "to contribute to specify the explanatory theory of the partnership" (p13, line 24). It also to me, more consistent with the methods described and overall a more
---

	manageable and achievable goal, given the complexity of the intervention. The title would then be something like: "The role of the UHC Partnership in strengthening policy dialogue for health planning and financing: a protocol for developing a realist theory of change" There is no study date or duration, and this needs to be included. Please also include the intended cessation year of the program (currently only the commencement year is included) P3-lines18-28 – how do these steps align with the steps specified for realist evaluation (e.g by Pawson and Tilley), where the third step is refining the initial program theory? P6 – this text should be shortened, retaining only the key concepts that help to contextualise the paper for the reader. P6-Table 1 : there is a misquote from Lacouture et al. Please check row two of the table – “a mechanism is an element of reasoning or of an agent’s reaction...” should be “A mechanism is an element of reasoning and reactions of (an) individual or collective agent(s) in regard of the resources available in a given context to bring about changes ...” P6-Table 1: I am not sure that there is any value in having this material in a table. P7-line 55. The ‘partnership synergy’ mechanism sounds promising. However this term is introduced and then dropped, and the reader is left wondering why it was rejected as a ‘mechanism’ (the mechanisms in the Figure 1 don’t include this, or are these aspects of partnership synergy?). This needs clarification and the term ‘mechanism’ needs to be used consistently. P8, lines 11-33: this text describes seems to describe a process chain. How does this relate to the characteristics of a mechanism that are provided in Table 1. I wonder why this is a mechanism, and not other things mentioned in the process chain for e.g. sense of ownership. Consider presenting this information in a figure. P8-line 32: unclear. P12 – Table 4 – Currently this Table is of limited value in helping to clearly communicate what is to be done. I suggest that this Table should succinctly present a complete list of the data sources and methods to be used, and elements of information to be collected from each. This detail can then be removed from the text. P13 – line 14 – please explain what theoretical literature will be considered. P13 – overall, it seems to me that you are describing a study that will develop a rich realist theory of change – but you are not actually testing the theory (see overall comments, point 4) P13 – line 28 to p14 – end: these sections provide more detail than needed and could be shortened.
--	---

REVIEWER	Lawrence Doi University of Edinburgh, UK
REVIEW RETURNED	01-May-2018

GENERAL COMMENTS	This protocol paper is well described. The data collection methods are sound and in line with the realist evaluation approach adopted. However, there are few points which may require further clarification. 1. In P7, L13-25 the authors mentioned that three sources of information were used to develop an initial realist intervention theory of the partnership. However, it appears that the sources were more than three. I can see the following information sources:  • Documents produced by WHO • Meetings conducted with stakeholders • Observation of inter country meetings
---

	 • Semi-structured interviews with stakeholders • Scientific literature on policy dialogue • Workshop I think the authors should consider revising that whole paragraph. Also, they should be more explicit about the type of data collected from these sources. For example, they mentioned that several meetings were conducted with stakeholders. What kind of data were collected at these meetings? 2. What is the relationship between development of the initial intervention theory described in P7 and the three initial theoretical propositions that were developed in P11 with inspiration from Jagosh et al? What is the difference between these two terminologies in the context of this paper? Please clarify. 3. Minor point: the duration of WHO support in table 3, is this in years or months? Add the appropriate unit.
--	--

VERSION 1 – AUTHOR RESPONSE

Reviewer: 1

Reviewer Name: Gill Schierhout

Institution and Country: The Kirby Institute, University of New South Wales, Sydney, Australia

Please state any competing interests or state 'None declared': None declared

General comments

This is an ambitious and important study of the implementation of a large systems-based initiative. I

have a reservation in the way the realist approach is described here, and think this needs some clarification in the manuscript. I have two overall recommendations that I respectfully suggest that the authors consider.

We would like to thank the reviewer for taking the time to read and give constructive feedback on our manuscript.

We hope we adequately answered his interrogations regarding the method.

Would the authors please consider presenting the initial program theory in a CMO structure.

This will

align better with the realist framework as the approach underpinning this study.

The initial program theory encompasses two sub-theories:

- One where the Partnership is the 'intervention' leading to a policy dialogue which is fed by knowledge,

inclusive and participatory, and led by MoH;

- One where the policy dialogue is the 'intervention' leading to the formulation of robust and comprehensive health policies, strategies and plans, alignment of stakeholders, and strengthened leadership of Ministries of Health

Our realist evaluation focuses on both sub-theories, which means that - depending on the focus - mechanisms

are different and will be triggered by different contextual elements. Instead of modifying the original model, we

submitted two additional figures. They provide a more precise picture of both sub-theories, trying to fit them into a

CMO structure.

Currently it is not clear to this reader what the CMO configurations are that will be tested and refined. If it

is not possible to do this, I would suggest that the research being conducted is 'theory development'

rather than 'theory testing' (see point 4 below).

According to our understanding of the approach, CMO configurations are part of the results of the study. They are

the first level of abstraction. This is why there are no CMO configurations in the protocol. There may be propositions 'about how mechanisms are fired in contexts to produce outcomes' (Pawson and Tilley, p.85). The objective is then to 'specify' those propositions through the identification of outcome patterns. Depending on the evaluation, such propositions are more or less developed. In our evaluation, developing such explanatory propositions is part of the study (step 1). As explained in the section 'Realist approach', a realist evaluation encompasses both theory development and theory testing, in an iterative process between 1) the conceptual and theoretical literature underlying the rationale behind the intervention, and 2) the empirical data coming from case studies. It is an explanation-building process, which is 'partly deductive (based on the statements or propositions at the outset of the case study) and partly inductive (based on the data from the case study). That is why we prefer not to label our study 'theory development' nor 'theory testing'.

We tried to make this more explicit in the manuscript.

I think the authors need to consider how the initial program theory will guide what kinds of data will be collected in the study. I wondered if the initial program theory as presented will be sufficient as a basis for a robust realist evaluation of a complex intervention in complex settings. If it is not, there needs to be a step in the study to develop such a program theory, which then forms the basis for the design of the evaluation. Currently it is not clear how the CMO configurations will emerge from the study.

*2
We thank the reviewer for his recommendation. Our realist evaluation adopts an iterative process whereby the program theory is refined alongside the research process, through an explanatory-building process. Thus, the study includes a step that aims at formulating explanatory propositions. To make this more explicit in the text, we provided additional explanations in Step 1 and Step 2 of the method.*

My understanding of realist evaluation is that it needs to build on learnings from previous programs that have used similar mechanisms (in possibly unrelated fields). This, in my understanding, the contribution of realism, but I don't see where this is included in the methodology. For example, theory from organisational change, or other disciplines.

In the new version of the manuscript, we explained the contribution of the theoretical literature that will be used to guide both the development of explanatory propositions and their specification.

I strongly recommend that the authors please consider revising the goal of the paper to be about "developing a realist theory of change" rather than actually doing a realist evaluation. This is consistent with what the authors own statement about the intended contribution of their analysis, being "to contribute to specify the explanatory theory of the partnership" (p13, line 24). It also to me, more consistent with the methods described and overall a more manageable and achievable goal, given the complexity of the intervention. The title would then be something like: "The role of the UHC Partnership in strengthening policy dialogue for health planning and financing: a protocol for developing a realist theory

of change”

We thank the reviewer for his recommendation. As explained previously, our realist evaluation adopts an iterative

process whereby the program theory is built alongside the research process.

There is no study date or duration, and this needs to be included. Please also include the intended

cessation year of the program (currently only the commencement year is included).

We included the duration of the study. As far as the program is concerned, there is no cessation year.

P3-lines 18-28 – how do these steps align with the steps specified for realist evaluation (e.g by Pawson

and Tilley), where the third step is refining the initial program theory?

We rephrased this section of the abstract so that it aligns with the study steps in the manuscript.

P6 – this text should be shortened, retaining only the key concepts that help to contextualise the paper

for the reader.

We removed the Table 1 to shorten the section, as well as some information less crucial for the reader.

P6-Table 1 : there is a misquote from Lacouture et al. Please check row two of the table – “a mechanism

is an element of reasoning or of an agent’s reaction...” should be “A mechanism is an element of

reasoning and reactions of (an) individual or collective agent(s) in regard of the resources available in a

given context to bring about changes ...”

We removed the table according to Reviewer’s comment (see below).

P6-Table 1: I am not sure that there is any value in having this material in a table.

We removed the table.

P7-line 55. The ‘partnership synergy’ mechanism sounds promising. However this term in introduced and

then dropped, and the reader is left wondering why it was rejected as a ‘mechanism’ (the mechanisms in

the Figure 1 don’t include this, or are these aspects of partnership synergy?). This needs clarification

and the term ‘mechanism’ needs to be used consistently.

At this stage, we were able to incorporate in the protocol additional elements to propose two sub-theories in the

form of context-mechanism-outcome. The partnership synergy literature becomes one of the theoretical sources

that support these two sub-theories.

P8, lines 11-33: this text describes seems to describe a process chain. How does this relate to the

characteristics of a mechanism that are provided in Table 1. I wonder why this is a mechanism, and not

other things mentioned in the process chain for e.g. sense of ownership. Consider presenting this

information in a figure.

This paragraph is the narrative explanation of Figure 1. Sense of ownership (along with others) is – as the

reviewer correctly points out and as reminded in Figure 1 – a mechanism. We added the letter ‘M’ after each

mechanism in the text. We did the same for Context (C) and Outcome (O). We moved this section in the method

section to better reflect the CMO propositions.

P8-line 32: unclear.

The sentence was removed.

3

P12 – Table 4 – Currently this Table is of limited value in helping to clearly communicate what is to be

done. I suggest that this Table should succinctly present a complete list of the data sources and methods

to be used, and elements of information to be collected from each. This detail can then be removed from the text.

We modified the table so as to answer the reviewer's concerns.

P13 – line 14 – please explain what theoretical literature will be considered.

We explained this further in the section 'Developing working propositions'. Theoretical literature refers to the

following main topics: policy dialogue, capacity building, knowledge translation, participatory and collaborative

approaches (from which the partnership synergy theory is derived).

P13 – overall, it seems to me that you are describing a study that will develop a rich realist theory of

change – but you are not actually testing the theory (see overall comments, point 4)

The reviewer is right: our attempt is to build an explanatory theory. The main output of our realist evaluation is the

building of a middle-range theory, and not a theory of change (which is not so different, except that a theory of

change may not be middle-ranged and may not use a realist framework). The testing of the theory is part of the

building of the theory.

P13 – line 28 to p14 – end: these sections provide more detail than needed and could be shortened.

We shortened the sections.

4

Reviewer: 2

Reviewer Name: Lawrence Doi

Institution and Country: University of Edinburgh, UK

Please state any competing interests or state 'None declared': None declared

This protocol paper is well described. The data collection methods are sound and in line with the realist

evaluation approach adopted. However, there are few points which may require further clarification.

In P7, L13-25 the authors mentioned that three sources of information were used to develop an initial

realist intervention theory of the partnership. However, it appears that the sources were more than three. I

can see the following information sources:

§ Documents produced by WHO

§ Meetings conducted with stakeholders

§ Observation of inter country meetings

§ Semi-structured interviews with stakeholders

§ Scientific literature on policy dialogue

§ Workshop

I think the authors should consider revising that whole paragraph.

We thank the reviewer for his suggestions. We slightly modified the paragraph so that the number of sources of

information used is correct. We also modified table 4 to summarize the steps of the study, methods used, and

status.

Also, they should be more explicit about the type of data collected from these sources. For example, they

mentioned that several meetings were conducted with stakeholders. What kind of data were collected at

these meetings?

Meetings were meant to discuss the underlying reasoning of the program planners for supporting policy dialogue

and for the type of support they provided, as well as how they thought their support would contribute to expected

outcomes. The data was qualitative in nature.

What is the relationship between development of the initial intervention theory described in P7 and the three initial theoretical propositions that were developed in P11 with inspiration from Jagosh et al? What

is the difference between these two terminologies in the context of this paper? Please clarify.

Following comments from Reviewer 1 and given the advancement of the study, we were able to provide additional information regarding the development of CMO propositions. The partnership synergy literature becomes one of the theoretical sources that support these two sub-theories.

Minor point: the duration of WHO support in table 3, is this in years or months? Add the appropriate unit.

We thank the reviewer for pointing out the potential misunderstanding of the numbers in the column. The numbers represent a scale from 1 to 3. We explain this in a note at the bottom of the table.

VERSION 2 – REVIEW

REVIEWER	Gill Schierhout The Kirby Institute for Infection and Immunity, UNSW Sydney, Australia
REVIEW RETURNED	03-Sep-2018

GENERAL COMMENTS	The authors have thoroughly addressed the issues raised in my earlier review. Thank you very much. I have no additional concerns.
---

REVIEWER	Lawrence Doi University of Edinburgh, UK
REVIEW RETURNED	28-Aug-2018

GENERAL COMMENTS	I think the authors have done a nice job of responding to the reviews. Overall, the paper is substantially clearer and easier to understand. My only addition is that Table 2 is still unclear. If the numbers in the column represent a scale of 1 to 3, what does 4 represent? The duration for Togo in the table is 4 and not 3 as you mentioned in the example.
---

VERSION 2 – AUTHOR RESPONSE

The comments from Reviewer 2 reveal typos in Table 2. The scale is actually from 0 to 4. We made the necessary changes in the manuscript.